# Development of a Manual Measurement Device for Measuring Hallux Valgus Angle in Patients with Hallux Valgus

**DOI:** 10.3390/ijerph19159108

**Published:** 2022-07-26

**Authors:** Guoli Li, Jinsong Shen, Edward Smith, Chetna Patel

**Affiliations:** 1School of Art and Design, Guangzhou Panyu Polytechnic, Guangzhou 511483, China; 2Textile Engineering and Materials Research Group, School of Fashion and Textiles, De Montfort University, Leicester LE1 9BH, UK; esmith@dmu.ac.uk; 3The Maths Learning Centre, De Montfort University, Leicester LE1 9BH, UK; chetnaptl@googlemail.com

**Keywords:** radiograph measurement, manual measurement, hallux valgus angular measurement, manual measurement device, reliability

## Abstract

Background: Hallux valgus (HV) is one of the most common forefoot deformities, and its prevalence increases with age. HV has been associated with poor foot function, difficulty in fitting footwear and poor health-related quality of life. The aims of this study were to design and develop an easy-to-use measurement device for measuring hallux valgus angle (HVA) in patients with HV and to assess the measurement reliability of the newly designed measurement device. Methods: A manual measurement device for measuring HVA was designed and developed to test on patients with HV. Two measuring methods, i.e., test–retest and intra-observer measurements, were used to evaluate the repeatability and reliability of the newly designed measurement device. In the test–retest measurements, a total of 42 feet from 26 patients with HV were repeatedly measured by the same researcher using the manual measurement device every 3 weeks over a period of 12 months. The measurement reliability of the newly designed measurement device was analysed based on the collected HVA data. In the intra-observer measurements, a total of 22 feet from the same group of HV patients were measured by the same researcher using the manual measurement device and by a consultant using X-ray measurement for comparison. The intraclass correlation coefficient (ICC) was used to determine the correlation of measurements between the manual measurement device and X-ray measurement. Results: The mean of the difference between the two repeat measurements of HVA using the newly designed manual device was 0.62°, and the average of ICC was 0.995, which indicates excellent reliability. The ICC between X-ray and the average of twice-repeated manual measurements was 0.868, with 95% CI (0.649, 0.947) (*p* = 0.000). When the relationship in HVA between X-ray measurement and manual measurement using the new device was regressed as a linear relationship, the regression equation was y = 1.13x − 4.76 (R^2^ = 0.70). Conclusions: The newly designed measurement device is easy to use, with low-cost and excellent reliability for HVA measurement, with the potential for use in clinical practice.

## 1. Introduction

Hallux valgus (HV) is an abnormal angulation and lateral deviation of the great toe at the first metatarsophalangeal (MTP) joint of the foot [1]. HV is the most common foot deformity and increases with age [2,3,4], with a prevalence of 23% in adults and 35.7% in elderly people [5,6]. Females are more likely to suffer from HV than men [1,7]. The ethology of HV is complex and multifactorial and might be caused by wearing improper footwear, abnormalities in foot anatomy and foot biomechanics, limb inequality, inflammatory joint diseases and genetic factors [8]. HV may lead to limitation of physical activities and foot pain [9,10].

In clinical practice, foot radiography can be performed via X-ray to assess precision foot disorders. From the radiographic image of the foot, the hallux valgus angle (HVA), the intermetatarsal angle (IMA) and the distal metatarsal articular angle (DMAA) can be measured to determine HV treatment for patients. HVA is the angle between the longitudinal axes of the first metatarsal and the proximal phalanx of the big toe. IMA is the angle between the longitudinal axes of the first and second metatarsals, whereas DMAA is the angle between the longitudinal axis of the first metatarsal and a line through the distal articular surface of the first metatarsal [11]. Surgeons consider all angles (HVA, IMA and DMAA) to assess the severity of HV in clinical practice if an X-ray photograph is available. However, among these angles, HVA is most commonly used as a predictor for assessment and correction of HV [12].

In recent years, many measuring devices have developed to measure HVA, including radiological measurement [13,14], smartphone measurement [11,15,16,17], footprint measurement [18] and digital photographs [19,20,21]. Radiological measurement is regarded as an accurate and commonly used tool in diagnosis and identification of the severity of HV. Radiological measurements are regularly used in clinical settings. Although a negligible dose of radiation is used for X-ray imaging, patients might still be concerned about the side effects of radiation exposure. Many specialized organizations have published recommendations suggesting limit of doses of radiation from X-ray in order to protect patients [22]. Munuera-Martínez et al. developed a simple instrument for the measurement of first ray mobility and demonstrated its concordance with radiographic measurement in order to diminish the use of X-ray imaging [23].

For smartphone measurements, a digital camera was used to take self-photographs of the feet when standing in a plantigrade position, and the images of the feet were analysed by the image-analysis software to determine angular deformity, such as hallux angles. It was found that the reliability of photographic HVA measurement was sufficiently high, but the reliability of photographic HVA measurement for repeated self-photography trials was lower, possibly due to the quality of photographs (i.e., blurriness and distortion), the positioning or the angle of the cameras towards the feet. [24,25].

Footprint measurement is based on the static footprint obtained by participants standing barefoot in a 398 × 312 × 191 mm scanning area. Based on the outline of the footprint, a hind–forefoot straight line is drawn from the medial border of the heel and tangentially to the ball of the big toe, whereas the forefoot–hallux line is drawn from the ball of the big toe to the medial border of the soft tissue of the big toe. Then, the HVA is calculated from the angle between the straight extension of the hind–forefoot line and the forefoot–hallux line [18].

In current clinical practice, there is a lack of a simple, easy-to-use and low-cost measurement device for measuring HVA. The Manchester scale [26] can be used to estimate HV severity by comparing the appearance of a patient’s foot with standardised photographs of four types of HV: no HV, mild HV, moderate HV and severe HV. However, the Manchester scale cannot provide exact HV angles.

Alternative devices are available to measure the foot’s length and width. A Brannock device is used to measure the foot’s length, width and arch length, whereas a Scriber block can be used to trace around the foot, although both of devices have difficulty in measuring HV feet. Therefore, the purposes of the current research were to design and develop a simple, easy-to-use measurement device for measuring HVA manually. A new method for HVA measurement using the newly designed device was evaluated to assess the reliability of the measured HVA through trials and practice on a group of HV patients, in addition to comparing HVA measurements between the newly designed device and X-ray measurements.

## 2. Materials and Methods

### 2.1. Study Design

The aim of the current study was to develop a new manual measuring device for measuring HVA in clinical practice, patients’ homes or care homes. In order to evaluate the measurement reliability of the new device, testing trials of HVA measurement were carried out on a total of 26 patients who suffered from mild or moderate HV deformity. The collected HVA data were used to assess the reliability of the newly designed manual device through test–retest method and statistical analysis. Intra-observer measurements were carried out by measuring the HVA of the same group of HV patients using the new manual measurement device and X-ray to evaluate their concordance.

### 2.2. Design of HVA Measurement Device

A Scriber block is usually applied to trace normal foot shapes. However, this device cannot trace deformed shapes because the side panel of a Scriber block is too wide and thick. Therefore, a new measurement device based on a Scriber block was designed with a curved shape that can trace the deformed foot and measure the HVA of patients.

Figure 1 shows the prototype of the designed measurement device and its actual dimensions. The width of the device at the bottom is 67 mm, and the height is 52 mm. The thickness of the device is 19 mm with a curved shape rather than straight shape, which can trace the deformed outline shape of an HV foot.

This new measurement device has advantageous features. First, the shape of the device is curved and smoothed-out, allowing for accurate and flexible movement, especially around deformed foot shapes. Secondly, the narrow front and sides of the new measurement device allow it to be held securely in order to follow the curved shape of the foot. Importantly, the pen is positioned slightly vertically so that the tip of the pen can be controlled in an accurate position. Therefore, the features of the device can achieve a reduction in errors during measurement. 

### 2.3. Measuring Method for Determining HVA Using the Newly Designed Measurement Device

The new measurement device was used to determine HVA. To improve the reliability of HVA measurement, a standard measuring procedure was set up as follows:(a)Participants sat barefoot, keeping a straight back on the chair. The chair was adjusted to ensure that the lower leg was vertical and at 90° relative to the upper leg. The feet were placed flat on the floor without bearing body weight (Figure 2a).(b)A piece of A4 blank paper was placed underneath the foot of the participant. A new tool, called the right-angle wooden ruler, which was designed by the researcher and made by a carpenter, was used to keep the foot steady and to ensure that the heel and lateral foot shape were positioned at a 90° angle. The outside of the foot and the heel were placed as close as possible to the right-angle wooden ruler (Figure 2b).(c)Three prominent points on the foot were located: the metatarsal phalange joint, the interphalangeal joint and the navicular. These three points were marked using a cross respectively on the foot to be measured (Figure 2c).(d)The new measurement device was used to trace the outline of the foot from the heel to the end of the big toe (Figure 2d). The foot shape outline was traced on the A4 paper.(e)A set square was used to record each of three prominent points (the medial border of the soft tissue of the big toe, the ball of the big toe and the medial border of the heel) on the A4 paper (Figure 2e).(f)A traced outline of the foot marked with the three prominent points was obtained on the A4 paper (Figure 2f).(g)A ruler was used to draw two tangent lines by connecting the most prominent points between the metatarsal phalange joint and the navicular, as well as between the metatarsal phalange joint and the phalanges joint. Two tangent lines were extended to meet each other (Figure 2g).(h)Finally, a protractor was used to record the HVA (Figure 2h).

All measurements of HVA were carried out by the same researcher in the current study.

### 2.4. X-ray Measurement to Determine HVA

X-ray photographs can provide a detailed structure of the foot to examine the three important radiographic parameters: HVA, IMA and HVI (the hallux valgus interphalangeal angle), as shown in Figure 3. In the current research, X-ray photographs were captured by a radiographic scanner (Yuwell DR 60); then, a consultant defined the longitudinal axis of the foot and the dashed lines on the digital image, as shown in Figure 3. The HVA was obtained from the X-ray photographs using measurement software.

### 2.5. Subjects and Criteria

Ethical approval for measuring HVA of patients in the current study was obtained from the Faculty Research Ethics Committee of De Montfort University, Leicester, UK. Patients diagnosed with HV were recruited to participate in this study. The criteria for participants to take part in the measurement trials included an age of 18 or older and suffering from mild or moderate HV deformity. The exclusion criteria were previous bunion surgery, pregnancy, bone fractures and diabetes. The Manchester scale [26] was used to identify whether participants suffered from HV. A total of 32 patients and 51 feet with HV aged between 20 and 72 years participated in this study. All patients were provided with oral and written information about the study and were required to provide written consent to take part in the study. 

### 2.6. Test–Retest Reliability

Test–retest reliability is the repeated measurement taken by a single person or instrument of the same item and under the same conditions. The reliability of the measurements can be improved by precise techniques and by carefully performing measurements at least twice to obtain an average measurement. Therefore, HVA was measured twice to obtain an average, each time using the newly designed measurement device. The difference between test–retest measurements of each HV foot was calculated. The mean of the difference of test–retest measurements of the HVA over 42 HV feet from 26 patients from day 1 to day 358 was obtained for statistical analysis.

### 2.7. Intra-Observer Measurements: X-ray and the Newly Designed Device 

In order to assess the reliability of the new measurement device, intra-observer measurements were carried out by measuring the HVA of the same group of HV patients using both the new measurement device and X-ray for comparison. As X-ray measurement is costly, 22 out of 42 HV feet were measured to evaluate the correlation between the newly designed measurement device and X-ray measurement in the same group of patients at the same time. The HVAs from X-ray were measured based on the bone structure of the foot between the shaft axis of the first metatarsal and the proximal phalanx of the hallux, whereas the HVAs from the newly designed measurement device were based on the traced outlines of the foot through two tangent lines over the most prominent points. The differences in the HVA between the two measuring methods were investigated. 

### 2.8. Statistical Analysis

In the current study, test–retest reliability was performed using IBM SPSS statistics version 25 to evaluate the reliability of the newly designed measurement device. The intraclass correlation coefficient (ICC) was used to investigate the correlation between the HVA measured using X-ray and that obtained with the new manual measurement device. A 95% confidence interval of ICC and significance *p*-value were calculated. Regression was used to analyse the relationship between HVA manual measurement using the new measurement device and X-ray measurement. Statistical significance was set at *p* < 0.05. 

## 3. Results

During HVA measurement over 12 months, 6 of the 32 recruited patients dropped out due to personal reasons, such as pregnancy or health situation. Therefore, 26 patients (42 HV feet: bilateral *n* = 16; left foot *n* = 6; right foot *n* = 4) were manually measured using the measurement device twice at a time every 3 weeks from Day 1 to Day 358 to test the reliability of the device. The difference of two repeat measurements of HVA for each foot and the mean of the differences of two repeat measurements of HVA over 42 feet from day 1 to day 358 were calculated as shown in Table 1. 

During 12 months of the HVA measurements, 6 of the 26 patients occasionally dropped out because of personal reasons, so the number of HV feet measured on the different days varied from 42 to 32. 

The results show that the mean of the difference of two repeat measurements of HVA ranged between 0.35 (*p* = 0.000) and 0.88, with an average of 0.62 (*p* = 0.000). The ICC (intraclass correlation) value ranged between 0.990 (*p* = 0.000) and 0.997 (*p* = 0.000) (indicating excellent reliability), and the average ICC was 0.995 (*p* = 0.000). The average of the 95% confidence interval of ICC was between 0.967 (*p* = 0.000) (lower bound) and 0.998 (*p* = 0.000) (upper bound). Therefore, based on the results of ICC, the new measurement device in combination with the HV measuring method had excellent reliability for HVA measurement.

### Correlation between X-ray and Manual Measurement

The X-ray method for measuring HVA is regarded as a commonly used tool in diagnosis and identification of the severity of HV. The reliability of the newly designed manual measurement device was further assessed by comparing the HVA measurement results between the new manual measurement and X-ray measurement in the same group of HV patients. X-rays and manual measurements using the newly designed device were taken separately within 2 days to prevent bias.

A total of 22 of 42 HV feet among 26 patients with HV were measured at the same time within 2 days using the new measurement device and using X-ray to determine HVA. An X-ray was taken of each patient’s foot only once by a specially trained professional with 20 years of experience in diagnostic radiography. The comparison and correlation between the HVA measurements from the new measurement device and from X-ray were analysed.

Table 2 shows the HVA of each foot of the patients measured manually twice (M_1_ and M_2_) using the newly designed measurement device. The data of the HVA measured from the first manual measurement (M_1_) show that these patients had different levels of HV severity, ranging between 10° and 33.2°. The HVA measured from the second manual measurement (M_2_) also ranged between 11.0° and 33.2°. The average of the HVAs of 22 HV feet was 19.8° and 20.4° from first and second measurements, respectively. The mean of HVA (M) for each foot was obtained by averaging the HVAs from two repeat measurements.

The data of the HVA measured from X-ray showed that these patients had different levels of HV severity, ranging between 14.2° and 33.2°. The average of the HVAs of 22 HV feet was 21.8°. 

To examine how close the measurements of the two methods were, the ratio of X-rays versus manual measurement using the measurement device was calculated (Table 3). The results show that the ratio was in the range of 0.7 to 1.3, and the average of the ratio was 0.9. We found that the two measurements provided good correlated data of HVAs.

In the current study, the intraclass correlation coefficient (ICC) was used to investigate the correlation of HVA measurements between X-ray and the new measurement device in the same groups of patients. Table 3 shows that the ICC between X-ray and the average of twice-repeated manual measurements was 0.868 (indicating good reliability), with 95% CI (0.649, 0.947) (*p* = 0.000). The results indicate that the correlation between X-ray and manual measurement had good reliability.

Regression analysis was carried out to estimate the relationship of HVA measurements between X-ray and the new measurement device. Figure 4 shows the relationship of the HVA measurements between the X-ray method (Y-axis) and the new measurement device method (X-axis) in a scatterplot. When the relationship of the measurements between X-ray and the new measurement device was regressed as a linear relationship, the following regression equation was obtained: y = 1.13x − 4.76. The result indicates a moderate (but close to substantial) regression (R² = 0.70). Therefore, X-ray and manual measurements had a moderate (but close to substantial) linear relationship.

## 4. Discussion

The objective of this research was to design and develop a new measurement device that is easy to use, low-cost and controllable for tracing of deformed feet, especially for measuring HVA of HV patients. A prototype of the new measurement device was produced with an established measuring method for HVA measurement with accuracy and reliability, with the potential for use in clinical practice.

The newly designed measurement device was tested to measure the HVA of as many as 42 HV feet twice each time every 3 weeks over the course of 1 year. Test–retest measurement was used to analyse the repeatability and reliability of the measurement. The differences in HVA between the repeat measurements across 42 HV feet were small, with an average of 0.62. Statistical analysis using the intraclass correlation coefficient (ICC) [27] confirmed the excellent reliability of the measurement based on the 95% confidence interval of the ICC estimate, with the boundary between 0.967 (lower) and 0.998 (upper) and an average *p*-value of 0.000. The accurate measurement achieved with few errors can be attributed to the advantageous features of the measurement device, especially capability of flexible movement and the use of same examiner taking measurements. 

Compared with a previous study by Yamaguchi et al. [25] on non-radiographic measurement of HVA using self-photography, we found that the newly designed measurement device and measuring method achieved higher reliability and fewer errors. Despite the similarity for the HVA measurement between photography and contour of the foot, the current measurement method has the advantage of measurement of the rested foot in the sitting position rather than of the body-weight-bearing foot in the standing position during the measurement. This method could be used to measure the patients with knee problems or difficulty in standing steadily. Another advantage is that the newly designed measurement device is much less costly and more durable and easier to use in the practice compared with smartphones which require proper positioning of the camera and additional costly software or apps for processing. Patients can also measure and evaluate their HVA in any convenient place.

When developing the new measurement device for measuring HVA, researchers always compare their measuring methods with the accurate X-ray radiograph method in term of repeatability and reliability for validation. In the current study, we found that the HVA ratio of X-ray measurement versus manual measurement using the new device was 0.9, on average (Table 2). The results of statistical analysis show that the intra-class correlation coefficients (ICCs) between X-ray and an average of twice-repeated manual measurements were 0.868, with a 95% CI of 0.649, 0.947 (*p* = 0.000), which indicates that there is good correlation between X-ray and the manual measurement method. Based on previous reports by Awatani et al. [11] and Nix et al. [21] with respect to the smartphone or photographic measurement method, we also found that measurement using the newly designed device matched the smartphone measurement method in terms of ICC correlation to X-ray measurement. Therefore, the new measurement device achieved good reliability in measuring HVA. 

The relationship between X-ray measurement and the new measurement device was regressed as a linear relationship to obtain the following regression equation: y = 1.13x − 4.76, which indicates a moderate regression. Thus, the equation can be used to predict X-ray HVA from manual measurements using the newly designed measurement device for mild and moderate HV. The results could also be used to assess the severity of HVA in clinical practice. Therefore, the current measurement device is an easy-to-use and less costly alternative to the X-ray method for monitoring patient HVA either in clinical practice or in personal residences.

The current study is subject to some limitations. This study was part of research involving the measurement of patients’ HVA and frequent monitoring over the course of one year. Therefore, a total of only 42 feet from 26 patients with mild and moderate HVA between the ages of 20 and 72 years were tested in these trials, and interobserver reliability was not assessed. The other limitation of this study is that the newly developed measuring method cannot be extrapolated to severe HVA before further testing is carried out. In order to reduce bias, an independent clinician could be employed for a study of HVA measurement using the new measurement device. Due to the limited budget for the current project, the researcher received proper training and carried out the measurements following the same measuring method under defined conditions for the consistent measurement of HVA to ensure that HVA data were reliable and consistent.

## 5. Conclusions

In the present study, we designed and developed a new measurement device that is easy-to-use, low cost and controllable for tracing of deformed feet, especially for measuring HVA of HV patients. The ICC between X-ray and the average of twice-repeated manual measurements indicates that there was good concordance between the X-ray and manual methods. The relationship of HVA measured between X-ray and the new measurement device was regressed as a linear relationship, to obtain a regression equation to predict X-ray HVA from the manual measurement using the newly designed measurement device. Therefore, the new measurement device exhibited excellent reliability in measuring HVA and could be an easy-to-use and less costly alternative to the X-ray method for monitoring patient HVA either in clinical practice or in personal residences.

## Figures and Tables

**Figure 1 ijerph-19-09108-f001:**
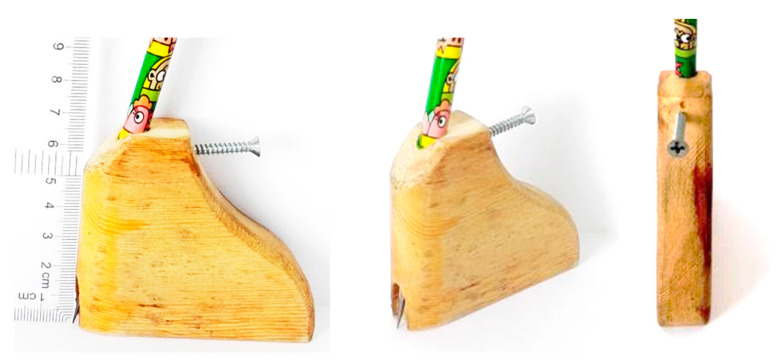
Design and characteristics of a newly designed measurement device.

**Figure 2 ijerph-19-09108-f002:**
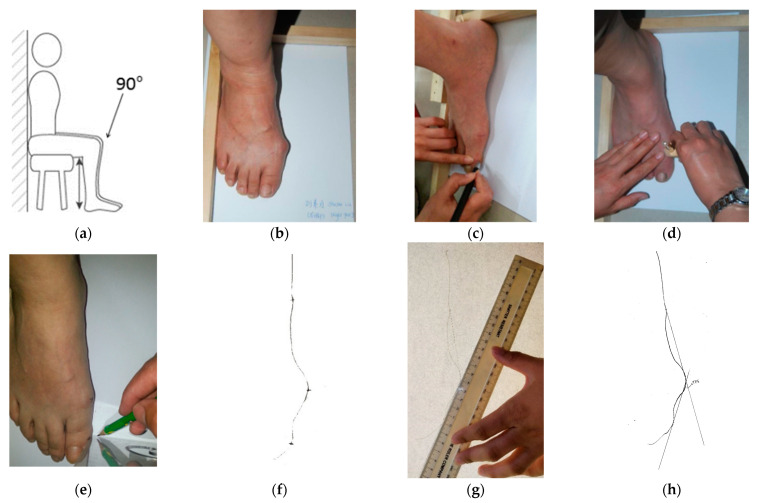
The operating procedure of HVA measurement: (**a**) sit vertically; (**b**) foot close to the corner of the right-angle frame; (**c**) three prominent points of the foot marked by a cross; (**d**) draw a line from the heel to the big toe; (**e**) vertical to mark on the line; (**f**) three prominent points reflected on the traced outline of the foot; (**g**) two tangent lines drawn; (**h**) a protractor used to measure HVA.

**Figure 3 ijerph-19-09108-f003:**
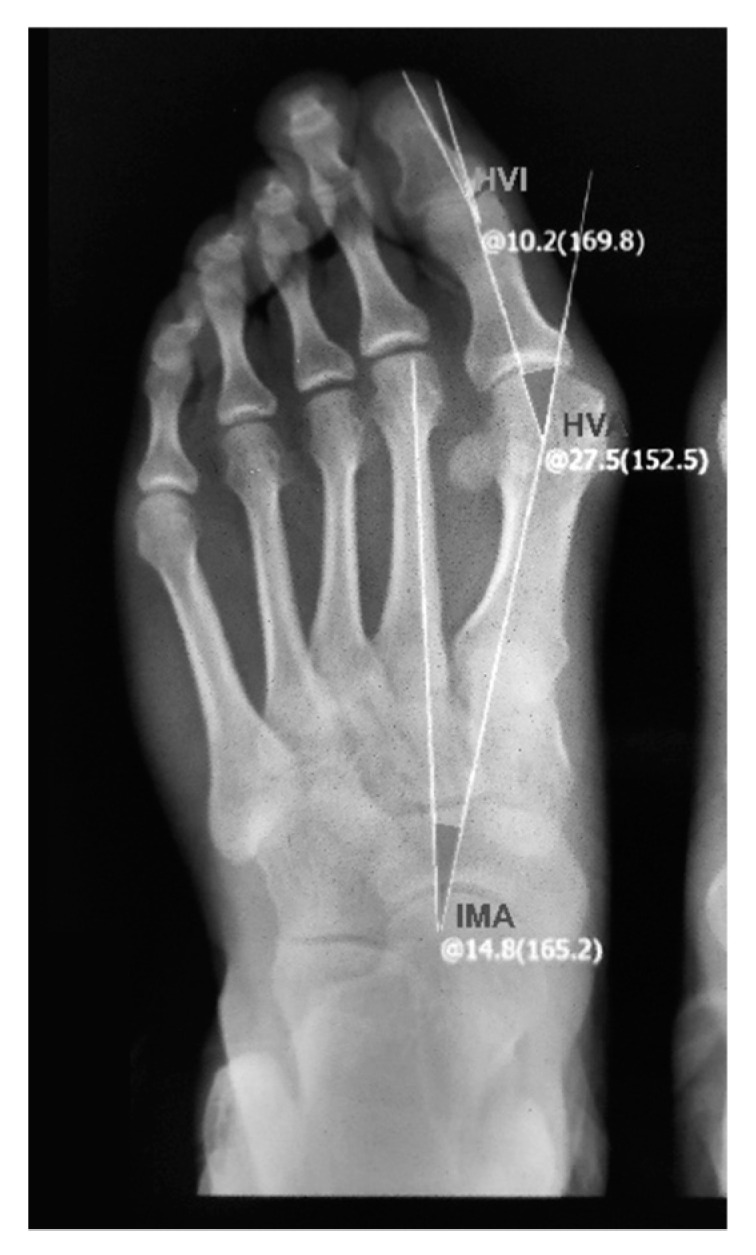
Radiograph of HV foot with radiographic parameters (HVA, IMA and HIA) captured by a radiographic scanner (Yuwell DR 60).

**Figure 4 ijerph-19-09108-f004:**
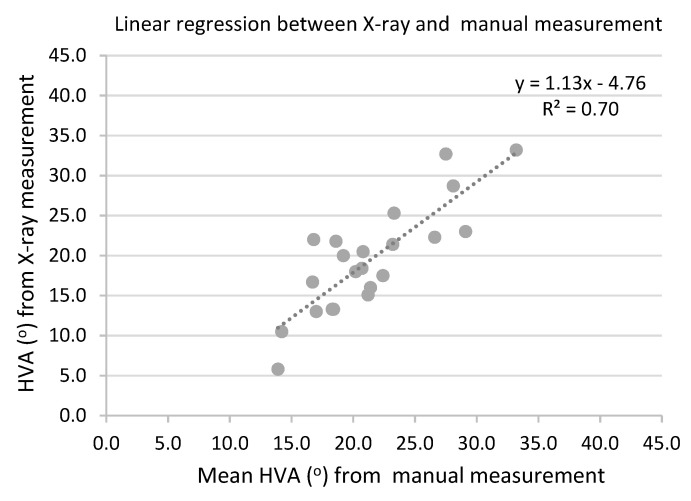
Regression between X-ray and manual measurements (*n* = 22).

**Table 1 ijerph-19-09108-t001:** Manual measurement and analysis using test–retested reliability and intraclass correlation coefficient (ICC).

Day	Number of Feet Measured Twice	Mean of the Difference of Two Repeat Measurements of HVA (°)	Intraclass Correlation (ICC)	95% Confidence Interval of ICC (95% CI)		Sig (*p*-Value)
				Lower Bound	Upper Bound	
D1	42	0.73	0.990	0.958	0.996	0.000
D22	42	0.46	0.993	0.985	0.997	0.000
D43	42	0.55	0.997	0.974	0.999	0.000
D64	42	0.45	0.997	0.984	0.999	0.000
D85	42	0.35	0.997	0.992	0.999	0.000
D106	42	0.67	0.990	0.971	0.996	0.000
D127	41	0.55	0.996	0.997	0.999	0.000
D148	41	0.65	0.994	0.971	0.998	0.000
D169	41	0.47	0.997	0.986	0.999	0.000
D190	39	0.49	0.997	0.980	0.999	0.000
D211	39	0.65	0.995	0.962	0.999	0.000
D232	39	0.74	0.994	0.935	0.998	0.000
D253	37	0.88	0.993	0.896	0.998	0.000
D274	35	0.58	0.995	0.978	0.998	0.000
D295	33	0.88	0.990	0.934	0.997	0.000
D316	32	0.67	0.994	0.968	0.998	0.000
D337	32	0.56	0.997	0.978	0.999	0.000
D358	32	0.78	0.995	0.951	0.998	0.000
Average	39	0.62	0.995	0.967	0.998	0.000

ICC: intraclass correlation coefficient; HVA: hallux valgus angle; D: day.

**Table 2 ijerph-19-09108-t002:** Differences between X-ray and manual measurements of HVA (*n* = 22).

	HVA (°)				
No.	M_1_	M_2_	M = (M_1_ + M_2_)/2	X-ray	Ratio (M: X-ray)
1	21.5	22.0	21.8	18.6	1.2
2	13.0	13.5	13.3	18.3	0.7
3	24.0	26.5	25.3	23.3	1.1
4	33.2	33.2	33.2	33.2	1.0
5	22.0	22.0	22.0	16.8	1.3
6	32.5	32.8	32.7	27.5	1.2
7	27.5	29.8	28.7	28.1	1.0
8	13.0	13.5	13.3	18.4	0.7
9	20.5	20.5	20.5	20.8	1.0
10	23.0	23.0	23.0	29.1	0.8
11	19.9	20.0	20.0	19.2	1.0
12	17.0	17.9	17.5	22.4	0.8
14	16.7	16.7	16.7	16.7	1.0
15	21.5	23.0	22.3	26.6	0.8
16	13.0	13.0	13.0	17.0	0.8
17	18.0	18.8	18.4	20.7	0.9
18	15.5	16.5	16.0	21.4	0.7
19	21.0	21.8	21.4	23.2	0.9
20	18.0	18.0	18.0	20.2	0.9
21	15.0	15.1	15.1	21.2	0.7
22	10.0	11.0	10.5	14.2	0.7
Mean	19.8	20.4	20.1	21.8	0.9

M_1_: first manual measurement. M_2_: second manual measurement. M: average of twice-repeated manual measurements.

**Table 3 ijerph-19-09108-t003:** Intraclass correlation coefficient (ICC).

Pairs	Number of Feet	Intraclass Correlation (ICC)	95% Confidence Interval of ICC		Sig (*p* Value)
			Lower Bound	Upper Bound	
X-ray vs. average of twice-repeated measurements M = (M_1_ + M_2_)/2	22	0.868	0.649	0.947	0.000
X-ray vs. first manual measurement (M_1_)	22	0.855	0.580	0.944	0.000
X-ray vs. second manual measurement (M_2_)	22	0.876	0.691	0.949	0.000

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
