# Peer review of "Development of a Manual Measurement Device for Measuring Hallux Valgus Angle in Patients with Hallux Valgus"

_ijerph, 2022, doi:10.3390/ijerph19159108_

Round 1

Reviewer 1 Report

General

The current manuscript looks to identify a newly designed device that can measure hallux valgus angle in the foot. The researchers designed this device with a curved shape that allows for better couture to the foot. They then measured individuals with HV and compared them to the current gold standard of x-ray. While the study was well performed, there are changes that need to be made before this manuscript can be published.

COMMENTS

General

The manuscript will need some moderate English language formatting. There are issues with tense, wording, and punctuation.

Abstract:

Page 1, Line 16 and 18: You should use ‘test-retest’, not tested-retested.

Introduction

General

I feel like the introduction would benefit with more discussion about how HVA is determined. What angles determine someone to have HVA and do angle on x-rays correlate to severity (mild, moderate, etc…). Plus, how are angles measured on x-rays. Are they done through a software program, or on printed x-rays.

Page 2, Lines  68-72: Your description of the iPhone app is misleading. The study by Yamaguchi et al. uses a self-photograph, not an x-ray photograph.

Page 2, Lines 89-92: The sentence beginning with ‘A new measuring method…’ should be reworded. It was extremely confusing to understand.

Methods

Figure 1: I’m not entirely sure what I’m even looking at in this picture. Can you take a better picture or maybe add another picture at a different angle?

Figure 2c, 2d, 2g, 2h: Again, not great pictures. These especially show up dark.

Page 3, Line 98: Why was only mild and moderate HV deformity used in this study?

Discussion

You spend a lot of time discussing the cost savings of this new device, but never compare prices of what any of this cost? Specifically, in line 306, how is the device less costly than an iphone app?

Page 10, Lines 316-320: If the new device matched smartphone measurements, what’s the advantage?

Page 10, Lines 328-333: These are not limitations. Some of your limitations include, but are not limited to, you only measured mild and moderate HVA so you cannot extrapolate to severe HVA. You don’t report the age of your patients, so this could be a limiting factor. Is the plan for the device to be commercially sold or will people be able to design there own?

Reviewer 2 Report

Dear Dear colleagues, your article is interesting. I thank you for your work and I hope that you will consolidate your research with a more decisive HAV meter. Congratulations

Author Response

Many thanks for your time and valuable comments on our manuscript.

Thanks for your encouraging comments. We will continue our research to improve the HVA measurement system.

Reviewer 3 Report

The manuscript reports the results in terms of reliability of the measurement of the angle of hallux valgus by a new device, and its agreement with the measurement in radiography. This study may be relevant for those situations in which it is necessary to know the value of this angle assuming a certain margin of error, but without the need to perform an X-ray (as will be discussed later, it is necessary for the authors to specify what the implications are for clinical practice.

As a strong point of the study, I would highlight the search for alternatives means of measurement from indirect estimates with an acceptable margin of error depending on their usefulness. The most relevant weakness is the absence of the fundamentals to support the sample size.

He then made some comments.

Abstract

Introduction and the entire manuscript

References must be numbered in order of appearance in the text. The reference list should include the full title, as recommended by the ACS style guide. The entire "References" section should be reviewed.

Material and method

I don't think the studio “was to develop a newly manual measuring device for measuring HVA in the clinical practice, patient’s or care home” because it is already developed, and they present the study to see the test-retest reliability and the agreement with the x-ray.

Authors need to specify the fundamentals that support the sample size.

In line 130 it is indicated that the measurement is carried out in discharge, and subsequently indicate this situation as a strength, however, in my opinion it is a weakness, since, by not considering the body weight, the reactive forces of the floor are not considered either, and consequently the possible rotational or translational moments that may occur in the bone segments that alter to a greater or lesser extent the measurement of the angle. This would also occur in the realization and measurement of x-rays. Taking the measurements in load can be considered a factor closer to the function of the foot in the different situations of daily life.

Subparagraph (d) in line 140 is redundantly worded.

Results

Table 1 does not specify that the values in the third column are degrees (º).

When making the measurements a single researcher, the study does not contemplate the different aspects of reliability: repeatability (test-retest reliability), intra-observer concordance and inter-observer agreement.

The description of the angles in section 2.4 is redundant.

In the manuscript 3 angles are specified and measured, however, later it seems that the only one of these relevant is the angle of hallux valgus, could you explain this?

In line 227 non-indica which units of measurement are (0.35; 0.62).

Discussion

The sentence “Thus, the equation can be used to predict x-ray HVA from the manual measurement using the newly designed measurement device" is a risky statement depending on the purpose you want to give to the results of the measurements with this method. So, it would be interesting to specify the uses of this measurement method beyond the graduation of the severity of the HV, and in the case of influencing this use, it would be advisable to discuss the advantages it can bring over other methods already described as the Manchester scale.

As limitations of the work, it would be interesting to discuss the number of subjects in both the reliability and concordance studies. It should also be considered that the assessment of inter-observer reliability has not been carried out. Finally, the possible implications for clinical practice of this measurement method should be clarified and specified.

References

It is necessary to review the style of citation of the references according to the indications of the journal.

Reference 18 is repeated.

Round 2

Reviewer 3 Report

Dear Authors,

I very much appreciate the effort made.

I would like to make some appreciations about the comments and changes made, as well as some aspects to consider. 

The abstract must be revised, according to the changes made throughout the manuscript, for example, the previous objective remains, not the new modification.

Keywords: I think the word “reliability” should be added (it is part of the main objective of the study)

Material and Methods: It is still not specified in the section of material and methods how the calculation of the sample size has been carried out.

The authors’ responses: authors think that the current measurement method might have the advantage from the measurement on the rested foot at the sitting position rather than on the body-weight-bearing foot at the standing position during the measurement. This could be used to measure the patients with knee problem or difficulty for standing steadily. The measurement could be affected less by the overweight of the body. This discussion was added in the line 301-304.

I still do not see the advantage of taking the measurement in discharge, since the value in load can be a more approximate value of what happens in the activities of daily living such as walking, where, of course, the overweight of the body is acting. Perhaps a discharge measurement could underestimate the results or deviate from the actual values during ambulation, where it is affected, for example, with one's own body weight. In any case, it is an issue that must be addressed, at least in the discussion, and not presented as an advantage, since, without objective data, it could be both an advantage and a limitation.

The authors’ responses: There is a lack of unified criteria for the classification of HV. Many studies classified HV according to hallux valgus angle (HVA). That is why HVA was used for measurement in this study.

I still do not understand why the other measurements (IMA and HIA) appear, which apparently do not provide relevant information to this study.

Line 325 et seq. I am still not clear about what advantage this method of measurement brings with respect to the graduation with the Manchester scale. (Line 325 et seq.)

It would be interesting to indicate why the evaluation of the inter-observer reliability was not carried out.
